# Cryogenic printing of three-dimensional soft iontronic sensors

Jinhao Li, Jie Cao, Yi Zhao, and Guoying Gu*, *Member, IEEE*

*Abstract—* **Designing soft iontronic skin for tactile sensing facilitates natural human-machine communications. However, the nonlinear characteristics of electrical transductions and mechanics usually compromise precise tactile decoding, requiring tailored three-dimensional architectures for responses. Herein, we propose a multimaterial cryogenic printing (MCP) technique that harnesses a universal all-in-cryogenic solvent phase transition strategy. We, therefore, can facilely fabricate sophisticated hydrogel structures with high aspect ratio geometries in high fidelity. Using this approach, we present a soft 3D-structured pressure sensor featuring a PEDOT:PSS-PVA hydrogel lattice encapsulated within an origami-inspired elastomeric framework. Our 3D lattice sensor enables wide-range linearity in both electrical responses ($R^2$ = 0.993±0.005 within 0-220 kPa) and compressive mechanics (~49.5% strain) without sacrificing tissue-like compliance. Further integration as human-machine interfaces facilitates accurate pressure input for wearable robotic teleoperation and an intelligent fingertip for safely detecting soft tissue modulus.**

## I. INTRODUCTION

Designing soft sensors to mimic biological skin facilitates natural haptic communications in telerobotics, virtual reality, and prostheses[1]. So far, numerous transduction mechanisms, including piezoresistive, triboelectric, magnetic, and capacitive, have contributed to pressure sensing, achieving high sensitivity, low hysteresis, and mechanical robustness[2-4]. Among these, soft iontronics leverage the electrical double layer in elastic electronic-ionic contacts to transduce signals, greatly enhancing the signal-to-noise ratio in a cluttered environment[5].

Despite these advances, soft iontronic sensors usually exhibit nonlinear electrical responses and mechanics due to the saturation effects of sensitivity and deformation[6]. For example, conventional 2.5D-structured designs based on microstructures inevitably cause the strain-stiffening effects under compression[7]. This restriction complicates parameter calibration, dynamic monitoring, and data processing in decoding diverse tactile interactions for position-controlled machines or robots. 3D structural engineering offers a promising route to program compressive behaviors[8, 9]. However, current fabrication techniques generally struggle to construct complex three-dimensional hydrogel architectures, owing to the inherent softness. The coupled electro-mechanical behavior and limited fabrication

capabilities remain key challenges in designing soft iontronic sensors for tactile sensing.

Herein, we present an MCP fabrication paradigm for 3D-structured soft hydrogel sensors. By proposing an all-in-cryogenic solvent phase transition strategy, our MCP paradigm facilitates conductive hydrogels into geometrically complex 3D structures[10]. Thus, we design and fabricate PEDOT:PSS-PVA hydrogel lattices sandwiched by metal-plate fabrics within an origami-inspired elastomeric framework. The as-formed 3DL sensor enables wide-range linearity, rapid responses, and high-resolution detection under extreme loading[11]. Using as wearable human-machine interfaces (HMIs), we enable accurate pressure input for sophisticated control signals in robotic teleoperation, as well as an intelligent fingertip to safely detect soft tissue elastic

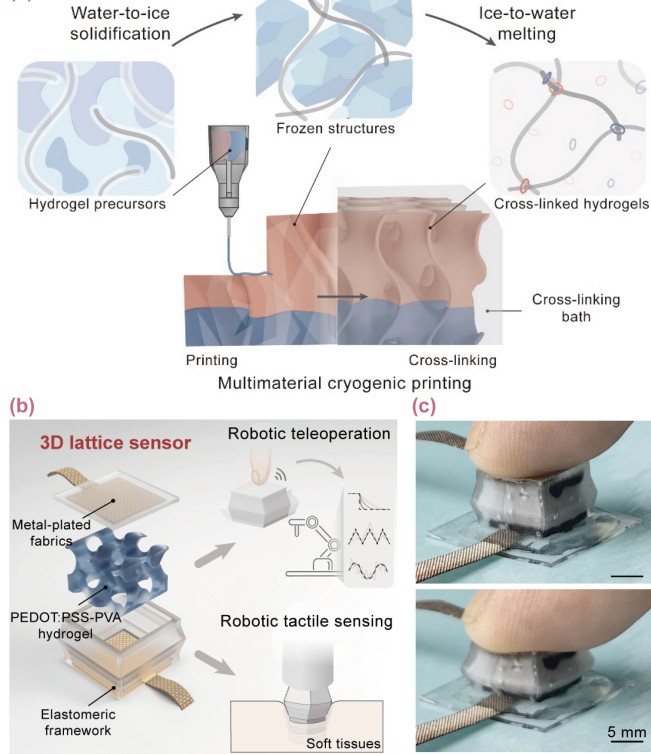

**Fig. 1 Cryogenic printing of three-dimensional soft iontronic sensors for decoding tactile interaction. (a)** Multimaterial cryogenic printing technique. **(b)** 3DL sensor for tactile interaction. **(c)** Photographs of operating a 3DL sensor.

## II. DESIGN AND FABRICATION

To eliminate nonlinear responses, we design PEDOT:PSS-PVA hydrogel lattices (8 × 8 × 6 mm) based on four gyroid units to trade off the fingertip-like sensor size (~1 cm) and high-resolution printed features (~100 µm). This hydrogel materials own the merits of mixed ionic and

*Corrosponding author. Mail: guguoying@sjtu.edu.cn.

1 State Key Laboratory of Mechanical System and Vibration, School of Mechanical Engineering, Shanghai Jiao Tong University;

2 Shanghai Key Laboratory of Intelligent Robotics, Meta Robotics Institute, Shanghai Jiao Tong University

This work was supported in part by the National Key R&D Program of China (Grant No. 2024YFB4707504), the National Natural Science Foundation of China (Grant No. 52025057, 524B2047), the Science and Technology Commission of Shanghai Municipality (Grant No. 24511103400) and Xplorer Prize.

electronic conductivity, ensuring stable signals and consistent responses under deformation. We further design origami-inspired elastomeric frameworks, where controllable folding deformations provide wide design spaces for linear compressive behaviors. Finally, we encapsulate metal-plate fabrics and sandwiched hydrogels inside silicone-based frameworks to construct our 3DL sensors. Combined with advanced printing techniques, our 3DL sensors have a size of 10 × 10 × 6.5 mm and exhibit mechanical reliability in repeating compressions (**Figure 1c**).

To fabricate the above hydrogel lattices, we design an all-in-cryogenic solvent phase transition strategy with two steps. The first step uses instant phase transition of water into ice to physically lock down the hydrogel precursors. By introducing a cryogenic platform into the direct-ink-writing system, we can rapidly solidify a variety of aqueous hydrogel inks into on-demand frozen 3D structures. The second step exploits reverse ice-to-water phase transition to initiate the chemical cross-linking of frozen hydrogel networks at the melting ice-water interface. To this end, we develop a cross-linker-containing bath with low freezing points to construct a cryogenic ice-water mixed system for cross-linking the immersed frozen structures. The diffusion of cross-linkers at the ice-water interface initiates synchronized ice melting and cross-linking reactions to transit pre-locked precursors into polymer networks. Such a strategy can effectively achieve free-standing yet complicated hydrogel lattices.

### III. Characterizations and Applications

To characterize the sensing mechanism, we in situ monitor the deformation behaviors using finite element analysis and X-ray computed tomography (**Figure 2a**). The results prove that the compression causes progressive collapsing of hydrogel lattices and a synchronous expansion of the contact area at hydrogel-electrode interfaces, thereby generating a growing capacitance signal. This mechanism facilitates wide-range electro-mechanical linearity. To demonstrate the sensing capability, we apply a tiny nut (0.15 g weight) on the 3DL sensor and observe an instant detection (~20 ms) containing clear rising and falling edges, exhibiting a limit of detection (LOD) of ~15 Pa. We also characterize the detection resolution of the sensor under high preloading (150 kPa), and the results show that even a slight stepwise loading (~1 kPa) can be clearly reflected in the capacitance signals

Robotic teleoperation through HMIs promises to augment human capabilities safely in hazardous or inaccessible environments. Typical demonstrations are usually achieved by comparing the amplitude of sensing signals with default thresholds to implement switch commands. It is still challenging to reproduce complex functions of traditional rigid controllers like impulsive control and proportional control. Herein, we preprogram the industrial controller to automatically identify the pressure input at HMIs into rapid click and sustained press, which remotely control the motion direction and feed velocity of an industrial robot, respectively (**Figure 2b**). Four movement directions were preprogrammed into four increments. A rapid click increments the selection through a predefined sequence, looping back after the fourth. Thus, the subject can teleoperate the robot to go through a planar labyrinth along the target trajectory with adaptive velocities, balancing rapid mobility and agile locomotion. This wearable teleoperation enables the integration with wristbands and gloves for natural interactions.

As another example, intelligent robotic fingertips are expected to identify physical properties of the touched objects through tactile interactions. However, typical non-linear electromechanical coupling behaviors of soft sensors usually require pre-trained machine-learning networks for information decoding. Herein, 3DL sensors enable safe and gentle touch for data collection and simplified processing for modulus detection (**Figure 2c**). By using a linear model, the soft intelligent fingertip enables accurate estimations with an average error of 8.9%, where its conformal contact allows rapid convergence in solving modulus under a biocompatible low pressure (<8 kPa). This soft, intelligent fingertip will enrich the tactile perception capabilities for position-controlled robots.

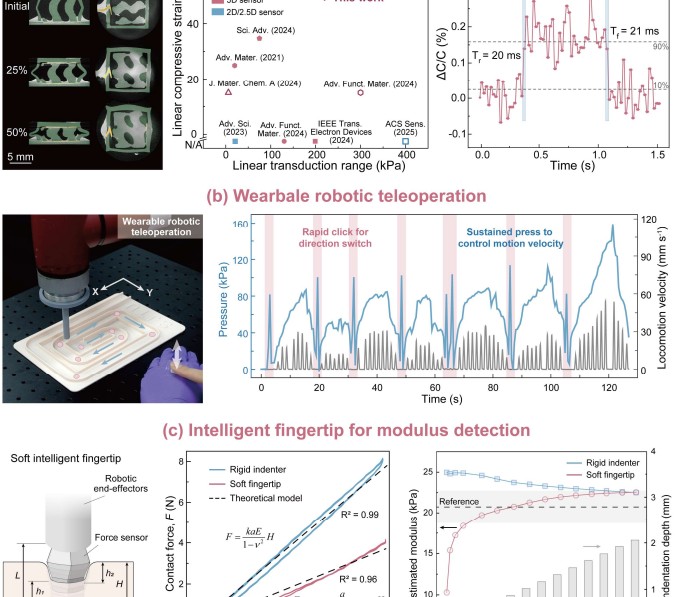

**Fig. 2 Performances and applications of 3DL sensors.** (a) Sensing mechanism and performances under compression (b) 3DL sensor for wearable robotic teleoperation (c) 3DL sensor as intelligent robotic fingertip.

### IV. Conclusion

In this study, we propose a cryogenic printing technique to enlarge material diversity and geometrical complexity for constructing 3D conductive hydrogel architectures. The as-developed soft 3D lattice iontronic sensor harnesses linear electro-mechanical behaviors for precise tactile sensing. Using as wearable HMIs, we enable accurate pressure input for sophisticated control signals in robotic teleoperation, as well as an intelligent fingertip to safely detect soft tissue elastic modulus. Our demonstrations explore broad prospects of 3D-structured hydrogels in soft sensors and robotics. Further improvements on consistency and durability will fuel the scaling of this technique for real-world applications.

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
