# OpenReview forum: "Cryogenic printing of three-dimensional soft iontronic sensors"
_IEEE.org/IROS/2025/Workshop/Tactile_Sensing — IROS 2025 Workshop Tactile Sensing OralPoster_

### Official Review · Reviewer_NYwc · 2025-09-19
**Review of Cryogenic printing of three-dimensional soft iontronic sensors**

**Rating:** 7
**Confidence:** 4

**Review:**

The paper presents a clear and comprehensive description of the design and performance of the sensor. I only have some minor comments. In the robotic teleoperation task, can the authors explain the options for commands in more detail? How did the direction switch operate, and can the user choose the direction? Additionally, although the task is interesting, what is the sensor's advantage compared to more conventional methods for robot control (e.g., simple buttons or a joystick)?

---

### Official Review · Reviewer_NWfw · 2025-09-21
**Review of Cryogenic printing of three-dimensional soft iontronic sensors**

**Rating:** 8
**Confidence:** 4

**Review:**

The manuscript presents a novel cryogenic printing technique for creating 3D soft iontronic sensors, focusing on the fabrication of hydrogel lattices for tactile sensing. The authors introduce an innovative cryogenic solvent phase transition strategy, which enables the high-precision creation of complex 3D hydrogel structures. The application of these sensors for robotic teleoperation and intelligent fingertip devices is effectively demonstrated. Overall, the manuscript is well-written, and the proposed method holds great potential for applications in soft robotics and human-machine interfaces.
To further strengthen the manuscript, it would be valuable to include a discussion on how this technique might be scaled up for practical use in real-world applications. Additionally, a comparison with existing soft sensors would help contextualize the advancements presented in this work.